# Adherence-Promoting Design Features in Pediatric Neurostimulators for ADHD Patients

**DOI:** 10.3390/bioengineering11050502

**Published:** 2024-05-17

**Authors:** William Delatte, Allyson Camp, Richard B. Kreider, Anthony Guiseppi-Elie

**Affiliations:** 1Center for Bioelectronics, Biosensors and Biochips (C3B^®^), Department of Biomedical Engineering, Texas A&M University, 400 Bizzell St., College Station, TX 77843, USA; william.delatte@tamu.edu (W.D.); allysoncamp@tamu.edu (A.C.); 2Exercise and Sport Nutrition Lab, Department of Health and Kinesiology, Texas A&M University, College Station, TX 77843, USA; rbkreider@tamu.edu; 3Department of Electrical and Computer Engineering, Texas A&M University, 400 Bizzell St., College Station, TX 77843, USA; 4Department of Cardiovascular Sciences, Houston Methodist Institute for Academic Medicine and Houston Methodist Research Institute, 6670 Bertner Ave., Houston, TX 77030, USA; 5ABTECH Scientific, Inc., Biotechnology Research Park, 800 East Leigh Street, Richmond, VA 23219, USA

**Keywords:** pediatric, adolescent, ADHD, internet-of-things, IoMT, neurostimulation, adherence, remote

## Abstract

The emergence of remote health monitoring and increased at-home care emphasizes the importance of patient adherence outside the clinical setting. This is particularly pertinent in the treatment of Attention Deficit Hyperactivity Disorder (ADHD) in pediatric patients, as the population inherently has difficulty remembering and initiating treatment tasks. Neurostimulation is an emerging treatment modality for pediatric ADHD and requires strict adherence to a treatment regimen to be followed in an at-home setting. Thus, to achieve the desired therapeutic effect, careful attention must be paid to design features that can passively promote and effectively monitor therapeutic adherence. This work describes instrumentation designed to support a clinical trial protocol that tests whether choice of color, or color itself, can statistically significantly increase adherence rates in pediatric ADHD patients in an extraclinical environment. This is made possible through the development and application of an internet-of-things approach in a remote adherence monitoring technology that can be implemented in forthcoming neurostimulation devices for pediatric patient use. This instrumentation requires minimal input from the user, is durable and resistant to physical damage, and provides accurate adherence data to parents and physicians, increasing assurance that neurostimulation devices are effective for at-home care.

## 1. Introduction

Health care delivery at home is emerging as a low-cost and efficacious alternative to in-clinic treatment [1]. Enabled by a host of emerging technologies that have given rise to the development of the internet of medical things (IoMT) [2], healthcare providers may now remotely monitor and provide therapeutic interventions for their patients [3]. The growth of IoMT applications in healthcare has established a new paradigm toward patient diagnosis, therapy, and monitoring in an extraclinical setting, although more work needs to be done to ensure this new pattern of healthcare delivery is both safe and effective for patients and straightforward for healthcare providers and insurers [4]. Clear benefits and increasing market penetration of IoMT-connected devices are key influencers that are expediting the growth of IoMT applications in healthcare [5]. IoMT implementations can benefit both healthcare providers and patients; remote health monitoring can improve patient outcomes and simplify chronic disease management while reducing costs and medical errors. IoT-connected devices are increasing in commonality, which, along with implementation of the appropriate guardrails for privacy in the management of medical data, bodes well to support future IoMT implementations and make them accessible to wider patient populations [6]. Public Wi-Fi hotspots in healthcare facilities are expected to triple from 2018 to 2023, making it the fastest-growing industry by this metric [7]. This will enable IoMT devices within healthcare facilities, such as hospitals, to function optimally. Additionally, it is expected that 50% of all networked devices will be IoT-enabled by 2023 [7]. These advances will simplify the implementation of IoMT devices for many different patient populations in many different healthcare delivery settings.

### 1.1. Neurostimulation Therapies for Pediatric ADHD Patients

A population with a particularly heightened need for IoMT technology implementation for at-home treatment modalities is the pediatric ADHD population for whom neurostimulation is indicated [8]. ADHD is a neuropsychiatric disorder that manifests in a pattern of inattention and/or hyperactivity-impulsivity, which interferes with daily life functions. The use of medical devices for the treatment of pediatric ADHD is an expanding sphere of research, with significant attention being paid to non-pharmacotherapeutic options for use in a home environment [9,10]. Neurostimulation has emerged as a key device-based, non-pharmacotherapeutic innovation for the treatment of ADHD in children [11] and adults [12]. Neurostimulation is the use of exogenously applied electric fields for the intentional modulation of the nervous system’s activity using invasive (e.g., deep brain stimulating microelectrodes) or non-invasive (e.g., transcranial stimulation) methods. Despite relatively small clinical trials to date, transcranial direct current stimulation (tDCS) and external trigeminal nerve stimulation (eTNS) represent quite promising interventions for the treatment of ADHD [8,13,14]. Both tDCS and eTNS methods involve application of external electrodes to specific anatomical locations on the patient, and use of an on-body pulse generator [8]. In eTNS, small electrical currents are delivered transcutaneously via supraorbital electrode(s) adhesively attached to the skin over the ophthalmic nerve [11]. When stimulated, electroceuticals, such as catecholamines, may be released that potentiate ADHD symptoms [14,15]. NeuroSigma (Los Angeles, CA, USA) was the first company to receive US Food and Drug Administration (FDA) clearance for a neurostimulation device with a pediatric ADHD indication, called the Monarch eTNS System^®^ [16,17,18]. While a promising intervention, attention must be paid to the practical implementation of such an intervention in an extraclinical pediatric ADHD patient population. 

### 1.2. Remote Adherence Monitoring for Investigation of Device Design Interventions

A central challenge presented by remote health care for children and adolescents with ADHD is that of adherence monitoring. When a patient is diagnosed, assessed, and treated in a clinical setting, the healthcare provider can readily monitor their adherence to therapy. However, when a patient is carrying out their treatment in an extraclinical setting, there is often no straightforward and accurate way for a healthcare provider to know the patient’s adherence patterns to the prescribed regimen. Additionally, there is minimal research on the development of adherence-encouraging design features for pediatric and adolescent ADHD patients. One possible intervention for adherence-encouraging design for medical devices is to offer patients a choice of color of the device, as well as offering the device in multiple colorways. A clinical trial protocol was designed to elucidate whether the choice of color, or a specific color, of a medical device could statistically significantly increase adherence rates in ADHD patients in an extraclinical environment. To understand adherence patterns of pediatric and adolescent patients with ADHD, an internet of things-based monitoring system was created for implementation in a neurostimulation device, called the Cerebro Monitoring System (CMS). The CMS is a sham device, not equipped with the capability to provide neurostimulation. The CMS is designed exclusively to enable the IoMT capability and to evaluate use pattern and user behavior.

## 2. Materials and Methods

### 2.1. Adherence Monitoring Using IoMT Headgear

An internet of things (IoT) approach was used to enable remote adherence monitoring of possible pediatric neurostimulation patients. The IoT, as applied in the medical field or Internet of Medical Things (IoMT), will vastly increase the collection, transmission, reception, and analysis of patient data [19]. Bluetooth Low Energy (BLE) was chosen for communication for this application because of its low power consumption requirements, reliability, and compatibility with commercially available mobile devices, such as the iPhone^®^. For this application, the mobile device acts exclusively as a central BLE device, while an Arduino board acts exclusively as a peripheral BLE device. The transmitted data was received through use of a commercially available iOS app, the LightBlue^®^ app (2023.1) by Punch Through, Inc., (Minneapolis, MN, USA), which enables a mobile device to scan for and be connected to peripheral BLE devices in the nearby vicinity, accept data from the peripheral device, and then send that data to the cloud via Adafruit IO [20]. The data was then stored with Adafruit IO for future retrieval [21].

### 2.2. Hardware Components

The CMS was designed to collect elapsed time of wear of each prototype by each patient. Additionally, the device was designed to signal donning and doffing. Each CMS consisted of a time-of-flight (ToF) sensor for measurement of the elapsed time since donning, a DelR Flex Sensor for signaling the donning or doffing of the device, an Arduino Nano 33 IoT board, and a battery holder. Each ToF sensor was purchased from Adafruit Industries (New York, NY, USA, Adafruit VL6180X), and uses a light detection and ranging (LIDAR) sensor to detect objects between 5 and 100 mm [22]. Each DelR Flex sensor is a 2 mm diameter, flexible, stretchable, carbon-filled, conductive rubber of 150 Ω/cm capable of 50–60% strain with a corresponding increase in resistance. Each Arduino Nano 33 IoT (“Arduino board”) was purchased from Arduino AG^TM^. Each Arduino board contains an on-board LSM6DS3 inertial measurement unit (IMU) with 3D gyroscope and 3D accelerometry capabilities and enables Bluetooth^®^ low energy (BLE) communication [23,24]. Figure 1 shows how the components were assembled to collect and send time of flight, gyroscope, and accelerometry data. Figure 2 is a set of photographs of the ToF sensor soldered in place via hook-up wires to the Arduino board.

### 2.3. Assembly of the CMS

The CMS device was powered via a micro-USB port on the Arduino board, as shown in Figure 2. Micro-USB battery holders containing three AA batteries were attached and used to power the CMS. All the hardware (the battery holder, Arduino board, ToF sensor, and DelR sensor) were held securely inside the fold of *NPJY* Unisex Beanie headwear purchased from Amazon. As shown in Figure 3, the CMS is a wearable soft fabric cap or “beanie” made of wool. Caps were neurostimulation shams fashioned in either of three colors: red (Pantone 200C ≈ AF060E/RGB: 175,6,14), blue (Pantone 7687C ≈ 263B83/RGB: 38,59,131), or gray (Pantone 424C ≈ 6D6D6B/RGB: 109,109,107). A small opening was cut on the inside layer of the fold to allow the ToF sensor to detect the distance from patient’s forehead. This opening was then made into a window, using clear transparent polyvinyl chloride (PVC) plastic sheet to create a comfort barrier between the patient’s skin and the ToF sensor. The 10 mm long (1.5 kΩ) DelR Flex sensor was similarly installed and secured within the fold of the beanie such that the sensor experienced a stretching force and corresponding strain (set to 10%) during donning and which force was relieved during doffing, producing a change of 150 Ω. In one approach, the ToF or DelR sensor was used to initiate and terminate data collection. In a second approach, data collection was initiated only when both sensors were activated. 

### 2.4. Clinical Trial Protocol Design

A clinical trial was designed for the testing of two hypotheses: (1) choice of color of a device will statistically significantly increase adherence rates in ADHD patients aged 8–12 in an extraclinical environment, and (2) a specific color of a device will not statistically significantly increase adherence rates in ADHD patients aged 8–12 in an extraclinical environment. Participants should be recruited and pre-screened before scheduling a familiarization visit.

### 2.5. Participant Pre-Screening

Pre-screening of participants includes limiting participants to children ages 8–12 who have a confirmed clinical ADHD diagnosis and full color perception abilities. Children and adolescents ages 8–12 were selected for study as the most popular segment for an ADHD-treating neurostimulation device. Children and adolescents with diagnosed ADHD will be required for this study because of the unique behavioral features that result from the symptoms of ADHD [25]. For example, children without ADHD may not have the same difficulty remembering or sustaining treatment tasks as would a participant with ADHD [25]. Thus, the largest improvement in adherence upon intervention is expected to be seen in children and adolescents with ADHD. Participants are also screened to only include those with access to mobile iOS^®^ devices to ensure compatibility with the CMS. ADHD, being a spectrum disorder, has resulted in a classification of patients as “mild”, “moderate”, or “severe” under the DSM-5 criteria. Our patient selection criteria do not control for the severity of the disorder. Similarly, patients and their parents/guardians may have differing motivations for participating in this study, which could include clinical manifestations, abhorrence of other therapies, etc. Our study design does not control for the above factors and assumes that each of these factors, severity of ADHD and motivation, is distributed equally among the various cohorts of those who were given or who select blue, red, or grey wearable devices.

### 2.6. Participant Familiarization Visit

After participants are screened and selected, they will be scheduled for a familiarization visit. During the familiarization visit, basic contact and demographic information are collected from each participant, as well as current medications, heart rate, and blood pressure. Participants in the study will then need to complete the ADHD Rating Scale-V to verify their clinical ADHD diagnosis [26]. Additionally, participants should be screened for colorblindness using the Ishihara Color Blindness Test to ensure they can distinguish between the colors of the devices [27]. Once it is verified that the participant meets all the inclusion criteria, assent is obtained from the pediatric participant and informed consent is obtained from the participant’s parent/guardian on behalf of their child. Once the assent form and informed consent forms are signed, the participant and their parent/guardian will be instructed on how to operate the device in detail, as well as given a take-home pamphlet for their future reference as needed. They will also be directed to the prescribed treatment (wearing) time of 20 min per day for 10 consecutive days.

### 2.7. Cohort Division

A total of 72 participants are to be recruited for this study. Each participant will then be assigned either the “choice” cohort or the “no-choice” cohort based on a “randomized” drawing. The “Choice” cohort participants (36) will be offered their selection from among the three CMS, one of each color, and will be asked to select the CMS of their choice, and then be given their selected device for use in the at-home study. Devices for the “no-choice” cohort participants (36) will be arbitrarily assigned, and they will not be made aware that there are other color options. This structure is illustrated in Figure 3. 

After the familiarization visit is completed and the participant completes their 10 days of sessions with the device, the participant will return to the testing facility for a final visit to return the device. This clinical trial protocol is only proposed as one possible option for investigating the two above hypotheses, and thus from here on, synthetic data, rather than patient data, is used to simulate the possible outcomes of such a study as discussed in the following sections.

### 2.8. Data Analysis Methodology

Data was compiled in Excel and all statistical analysis was conducted in JMP^®^ Pro 15.0.0 on a Mac. The synthetic CMS data were organized by participant and summed for a total elapsed treatment time per participant. These values, the total elapsed treatment times, were compared to the total prescribed treatment time of 200 min (20 min per day for 10 days). A simple equation (Equation (1)) was used for the calculation of an adherence rate of patients to the neurostimulation device.

Equation (1): Adherence rate equation
(1)$$AR=\frac{total\;elapsed\;treatment\;time\;(minutes)}{total\;prescribed\;treatment\;time\;(minutes)}*100$$

Two-factor analysis of variance (ANOVA) with replication was run on the total data set and analyzed for statistically significant sources of variation, with a null hypothesis and alternative hypothesis as shown in Equation (2). Alpha for all following tests was set to 0.05. 

Equation (2): ANOVA null hypothesis and alternative hypothesis.
(2)$$H_{0}\;:{µ}_{1}={µ}_{2}\;\mathrm{and}\;H_{a}\;:\;{µ}_{1}\neq {µ}_{2}$$

If the *p*-value obtained was less than alpha, the null hypothesis was rejected, and the source of variation was found to be statistically significant. If the *p*-value obtained was greater than alpha, the null hypothesis failed to be rejected, and the source of variation was found not to be statistically significant. If a source of variation was found to be statistically significant then unpaired two-tailed pairwise student’s *t*-tests were run to identify the statistically significant differences between groups, with a null hypothesis and alternative hypothesis as shown in Equation (3).

Equation (3): Student’s *t*-test null hypothesis and alternative hypothesis.
(3)$$t_{0}\;:{µ}_{1}={µ}_{2}\;\mathrm{and}\;t_{a}\;:\;{µ}_{1}\neq {µ}_{2}$$

Power analysis was conducted for each *t*-test to determine the number of participants needed to detect a given difference in the means. Equation (4) was used to determine the number of participants needed for a power of 80%.

Equation (4): Number of participants calculation using power and difference in the means.
(4)$$n=\sigma^{2}\frac{{\left(Z_{\alpha }+Z_{\beta }\right)}^{2}}{\Delta^{2}}$$

Input values included β = 0.2, difference in the means (Δ) = 0.1, and α = 0.05. Number of participants, *n*, was calculated for each model situation.

## 3. Results and Discussion

Recent US FDA approval of neurostimulation devices to address pediatric ADHD has resulted in increased attention to development and deployment of neurostimulation devices and systems. Proper progress necessitates extraclinical research regarding the in-clinic and/or at-home deployment of such therapies in a population that has difficulty remembering tasks and staying attentive to therapy. In related work, the authors have employed the Lean LaunchPad^®^ methodology to engage and report on the perceptions and contributions of stakeholders (clinician providers, parents, teachers, and adolescent patients with their parents or guardians) to the development and deployment of external neurostimulation devices for the treatment of ADHD. Stakeholders expressed a desire for new ADHD treatments to emphasize appeal to children to promote adherence and to include a remote adherence monitoring component in order to maintain strong evidence of efficacy. The present comparative effectiveness trial seeks evidence-based insight into the influence of color and choice of color in influencing pediatric patient adherence to neurostimulation therapy using sham hardware devices.

There are six dominant theories guiding thought and practice related to patient adherence to therapy [28,29]. A theoretical framework allows us to understand why patients may or may not follow their prescribed treatment plans [30]. The Health Belief Model (HBM) suggests that patient adherence is influenced by their beliefs about the severity of their condition, the likely benefits of treatment, the perceived barriers to treatment, and exogenous and endogenous cues to action [31]. If a patient perceives their illness as serious, believes in the effectiveness of the treatment, sees few obstacles to adherence, and receives prompting or reminders to adhere, they are more likely to comply with therapy. The Theory of Reasoned Action (TRA) and Theory of Planned Behavior (TPB) advances that patients are more likely to adhere to therapy if they have positive attitudes toward the treatment, perceive that other important persons support their adherence, and believe they can adhere despite challenges [32]. These theories emphasize the role of individual attitudes, subjective norms (social pressures), and perceived behavioral control in determining health behaviors, including adherence. The Social Cognitive Theory (SCT) posits that patient adherence is influenced by reciprocal determinism involving personal factors (e.g., beliefs, self-efficacy), behavioral factors (e.g., skills, habits), and environmental factors (e.g., social support, access to resources) [33]. Patients with higher self-efficacy, i.e., belief in their ability to carry out specific behaviors, are more likely to adhere to therapy. The Self-Regulation Theory focuses on patients’ self-monitoring, self-judgment, and self-reaction processes in managing their health behaviors [34]. Adherence is therefore influenced by the patients’ ability to set specific goals, monitor their progress, and adjust their behaviors based on feedback. The Transactional Model of Stress and Coping highlights the role of stress and coping mechanisms in patient adherence [35]. Patients facing high levels of stress due to illness or treatment side effects may be less likely to adhere. Coping strategies and social support thus play crucial roles in helping patients manage stress and maintain adherence. The Social Ecological Model (SEM) emphasizes the importance of broader social and environmental factors in influencing patient adherence [36]. These include family dynamics, community resources, healthcare access, and policy contexts. Patients are more likely to adhere to therapy when they have support from multiple levels of influence, including family, peers, healthcare providers, and community organizations. Understanding these theories helps providers develop more effective interventions to improve patient adherence. Each of these theories favor an empowered patient; such empowerment may manifest in simple matters such as preferred color and choice of a neurostimulation device.

### 3.1. Adherence Data Collection Using CMS

The IoMT headgear was developed to provide a first-stage prototype for remote adherence monitoring of a neurostimulation device. Components used include an Arduino Nano 33 IoT board, an Adafruit VL6180X time-of-flight sensor, a DelR Flex sensor, a micro-USB battery holder, and a mobile device containing the LightBlue^®^ BLE prototyping app and Adafruit IO Cloud Connect capability. These components, in coordination, can detect deployment of the device on the head of the patient, allow us to determine the initiation, termination, and elapsed time of a treatment session for a neurostimulation device. These data are then accessible via Adafruit IO Cloud Connect for remote researcher or physician viewing. The implementation of multiple measurement modalities in the CMS, ToF, and DelR Flex sensors increases assurance that the device is being worn by a human, as it provides that (i) a wearing surface is proximal to the device and (ii) that the device is being stretched into the wearing position. The differential change in resistance could eventually be designed to meet the specific head dimensions of the particular patient. Design features such as color and social factors such as choice empower the pediatric patient to adhere to therapy and contributes to overall user friendliness and acceptance.

### 3.2. CMS Software Specifications

Ethical considerations regarding privacy, consent, and the potential for over-surveillance are major concerns, particularly as it relates to pediatric patients. The CMS requires that a consenting parent or guardian engage a dual authentication initiation of the data collection and transmission system. The software of the CMS contains the code that controls the actions of the Arduino board, the ToF sensor, and the DelR sensor, as shown in the system’s functional flowchart in Figure 4 Functional flowchart for performance of the Cerebro Monitoring System (CMS). (A) When data acquisition is made conditional on either of the two sensors being activated. (B) When data acquisition is made conditional on both sensors being activated. After setup and initialization of the IMU, DelR Flex Sensor, ToF sensor, and BLE capabilities, the Arduino board advertises as a peripheral BLE device. Following connection confirmation with the central BLE device, two algorithms of event detection are proposed. In Figure 4A, the DelR Flex Sensor and IMU are read simultaneously. If either detection threshold of 150 Ω or 1.1 g-force units in any direction are exceeded, then data recording begins. The threshold of 1.1 g was determined based on experimental use and ensures that the following loop is only entered when the device is picked up and placed on the head, not simply shuffled around. In the alternate algorithm of Figure 4B, once a connection is initiated by a central BLE device, the Arduino board begins reading accelerometry data from the board’s IMU. If movement greater than 1.1 g-force units (g) in any direction is detected, then the DelR Flex Sensor is read to confirm donning of the device due to stretching in excess of 10% strain. If a resistance change of 150 Ω or greater is detected then the start time is recorded. In both algorithms, after the start time is recorded, the ToF sensor begins continuously reading range values. It continues to read range values until a range value of 50 mm or above is reached, indicating that the device has been removed from the head. The threshold of 50 mm was determined based on experimental use and ensures that the loop only breaks when the device is removed from the head and not when it is adjusted for fit or position. Once this threshold is reached, the device stops reading range values and records the stop time. The elapsed time is then calculated by subtracting the start time from the end time, and then the elapsed time (in milliseconds) is written to the connected central BLE device. This process can repeat as many times as a large movement is detected while still connected to the central BLE device, ensuring that if multiple sessions occur in sequence, they will all be detected separately and accurately. Figure 5 shows the graphic user interface (GUI) of the CMS App.

### 3.3. Clinical Trial Data Model 1: Statistical Analysis

Three models of trial outcomes were explored. The first, Model 1, establishes that color choice is a statistically significant variable in adherence rate and color is not a statistically significant variable in adherence rate. For this model, each group’s mean adherence rate is graphed and labeled in Figure 6. 

The ANOVA resulted in *p*-values of 4.999 × 10^−13^ and 0.865 for choice and color as sources of variation, respectively. Because the *p*-value for choice as a source of variation was less than α = 0.05, the ANOVA null hypothesis was rejected, and this was found to be a statistically significant source of variation in the adherence rate data. Because the *p*-value for color as a source of variation was greater than α = 0.05, the ANOVA null hypothesis failed to be rejected, and this was found not to be a statistically significant source of variation in the adherence rate data. Because choice was found to be a statistically significant source of variation in adherence rates in Model 1, an unpaired two-sided student’s *t*-test was conducted between the choice and no-choice groups. This *t*-test resulted in a *p*-value of 2.11 × 10^−12^. Because the *p*-value of this *t*-test was less than α = 0.05, the *t*-test null hypothesis was rejected, and this was found to be a statistically significant difference. Power analysis was conducted to elucidate the number of participants that would be needed per group to draw conclusions with at least 80% power (Equation (4)). The data used in the above *t*-test between the choice and no-choice groups indicated that at least FIVE (n = 4.53) participants per group (choice and no-choice) were needed for the test to be properly powered.

### 3.4. Clinical Trial Data Model 1: Conclusions

These model data indicate that choice is a statistically significant source of variation within the data, while color is not a statistically significant source of variation within the data. Further analysis via student’s *t*-test indicates that the adherence rate of the choice and no-choice groups are statistically significantly different. These findings reveal that it could be helpful to offer aesthetic choices of medical devices to pediatric ADHD patients as a measure to increase adherence rates to treatment. It does not explicitly matter which specific colorways are offered to patients in the measurement of adherence rates, just that there are multiple aesthetic options offered. Choice is empowering during the decision-making process. Children experience positive feelings when given choices; allowing a feeling of autonomy that establishes motivation during preferred, and more importantly non-preferred, activities.

### 3.5. Clinical Trial Data Model 2: Statistical Analysis and Conclusions

Model 2 represents one of the possible outcomes of the data: that neither color choice nor color is a statistically significant variable in adherence rate. For this model, each group’s mean adherence rate is graphed and labeled in Figure 7.

The ANOVA resulted in *p*-values of 0.215 and 0.388 for choice and color as sources of variation, respectively. Because the *p*-values for choice and color as sources of variation were both greater than α = 0.05, the ANOVA null hypothesis failed to be rejected and these were both found not to be a statistically significant source of variation in the adherence rate data. Because of this, no additional *t*-tests were performed, and no power analysis was completed. These findings indicate that other device features and environmental factors should be considered and researched as possible influencers to increase patient adherence rates in a pediatric ADHD setting. 

### 3.6. Clinical Trial Data Model 3: Statistical Analysis

Model 3 represents the final of the chosen possible outcomes of the data: that color choice is not a statistically significant variable in adherence rate, and that color is a statistically significant variable in adherence rate. For this model, each group’s mean adherence rate is graphed and labeled in Figure 8.

The ANOVA resulted in *p*-values of 0.36 and 5.25 × 10^−8^ for choice and color as sources of variation, respectively. Because the *p*-value for choice as a source of variation was greater than α = 0.05, the ANOVA null hypothesis failed to be rejected and this was found not to be a statistically significant source of variation in the adherence rate data. Because the *p*-value for color as a source of variation was less than α = 0.05, the ANOVA null hypothesis was rejected, and this was found to be a statistically significant source of variation in the adherence rate data. Because choice was found to be a statistically significant source of variation in adherence rates in Model 3, an unpaired two-sided student’s *t*-test was conducted between the choice and no-choice groups. This testing revealed *p*-values of 2.21 × 10^−3^, 9.46 × 10^−10^, and 3.46 × 10^−3^ for blue/grey, blue/red, and grey/red groups, respectively. Because the *p*-values of each of these *t*-tests were less than α = 0.05, the *t*-test null hypothesis was rejected for each pairwise combination, and they were all found to be statistically significant differences. Power analysis was conducted to elucidate the number of participants that would be needed per group to draw conclusions with at least 80% power. The power analysis yielded *n* values of 8(7.42), 10(9.76), and 9(8.03) for blue/grey, blue/red, and grey/red groups, respectively. These values indicate that at least ten participants per group (blue, grey, and red) are needed for the data to be properly powered.

### 3.7. Clinical Trial Data Model 3: Conclusions

These model data indicate that choice is not a statistically significant source of variation within the data, while color is a statistically significant source of variation within the data. Further analysis via student’s *t*-test indicates that the adherence rates of the color groups are statistically significantly different. These findings reveal that it could be helpful to offer medical devices in specific colorways (in this data, red colorways) to pediatric ADHD patients to increase adherence rates to treatment. Different colors are perceived to mean different things. For example, tones of red lead to feelings of arousal while tones of blue are often associated with feelings of relaxation. Both emotions are pleasant, so therefore the colors themselves can produce positive feelings that support adherence.

## 4. Conclusions

As neurostimulation treatment options for ADHD are being developed, patients are better able to understand and manage their symptoms. However, pediatric patient adherence to these new treatments is crucial, and with the current understanding of the symptoms of this diagnosis, this presents a challenge. The attendant challenge of patient adherence to therapy will necessitate design considerations to promote patient adherence, particularly in pediatric and adolescent populations. This includes ergonomic and aesthetic design considerations. The clinical trial protocol discussed is designed to elucidate the effects of color and color choice of a neurostimulation device on adherence rates for pediatric ADHD patients. The data discussed is simulated to model three possible scenarios as outcomes to the clinical trial and have different implications for future work. Model 1 (color choice is significant, color is not significant in influencing adherence rate) indicates that more work should be conducted to identify how color choice influences adherence rates, whether this hypothesis holds for additional populations and larger sample sizes, and if there is any relationship between the number of color choices and adherence rates. Model 2 (neither color choice nor color is significant in influencing adherence rate) demonstrates that more work should be conducted to identify other possible factors to influence adherence rates, such as awareness of monitoring technology, mobile reminders and notifications, or various form factors. Model 3 (color choice is not significant, color is significant in influencing adherence rate) signals that more work should be conducted to refine how color relates to adherence rate and which hues, tints, tones, and shades are most effective in influencing adherence rate. The findings of this study apply only to this study and may not extrapolate to other devices. However, studies such as this one should encourage others to look at medical device technology through the lens of such adherence factors. This is particularly relevant for spectrum disorders, where clinical outcomes are likely strongly influenced by measures of adherence to therapy. Overall, this clinical trial protocol is a powerful and relatively practical way to realize the possible effects of aesthetic design and aesthetic choice on adherence rates in a sample of pediatric ADHD patients. This protocol is made possible by the development of an accurate, remote, and ambient adherence monitoring system. 

## 5. Future Work

Future work could include the verification and validation of the Cerebro Monitoring System (CMS) as an accurate and robust tool for remotely measuring adherence rates. The code should also be made more efficient to reduce power consumption and enable the system to be used over longer periods of time. Future work could also include further miniaturization and the development of a sturdy and adaptable housing for the CMS components to ensure its integrity through longer periods of use. Additionally, the clinical trial could be carried out with a large sample size. This was out of the scope of this body of work due to time and funding constraints, as well as the ongoing COVID-19 pandemic, which greatly limited participant recruitment and engagement with a possible study. 

## Figures and Tables

**Figure 1 bioengineering-11-00502-f001:**
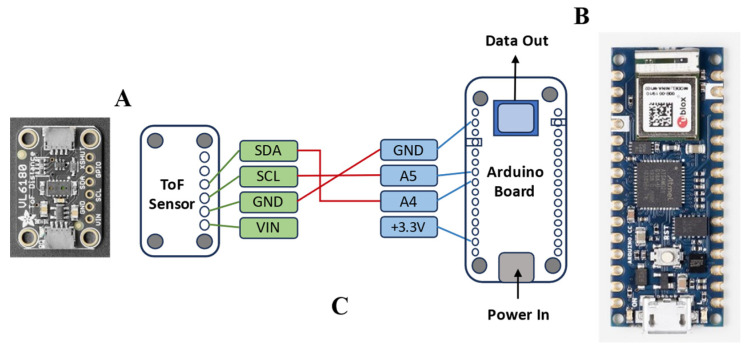
Images of the Adafruit VL6180X Time of Flight Distance Ranging Sensor (VL6180) (**A**), the Arduino Nano 33 IoT (**B**), and a schematic illustration of the assembled ToF sensor and Arduino board showing the pin-out diagram (**C**).

**Figure 2 bioengineering-11-00502-f002:**
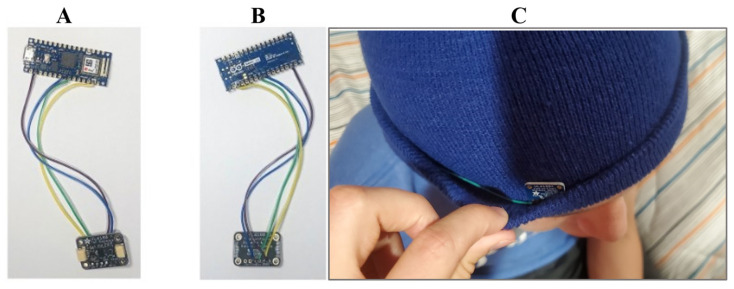
Images of both the front (**A**) and back (**B**) of the wiring hook-up connections between the ToF sensor and the Arduino board. (**C**) Photo of device incorporated into the cap.

**Figure 3 bioengineering-11-00502-f003:**
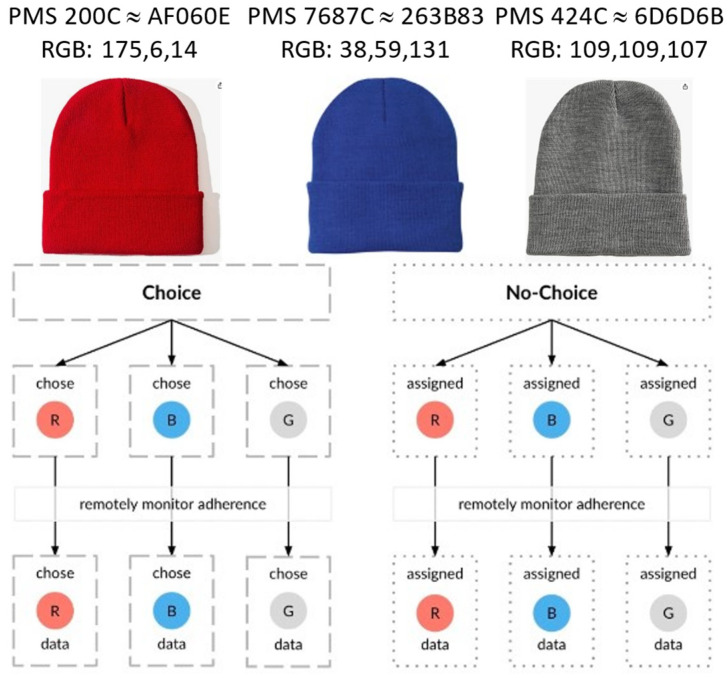
Colored headwear for instrumentation and proposed clinical trial cohort division of headwear color among the “choice” and “no-choice” assigned groups.

**Figure 4 bioengineering-11-00502-f004:**
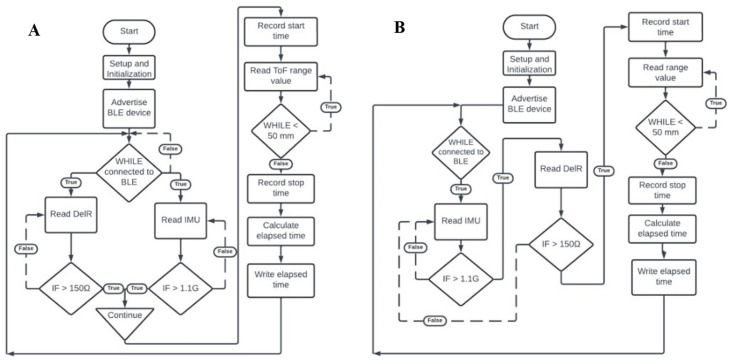
Functional flowchart for performance of the Cerebro Monitoring System (CMS). (**A**) When data acquisition is made conditional on either of the two sensors being activated. (**B**) When data acquisition is made conditional on both sensors being activated. In this application, there is one service containing one characteristic–elapsed time. Once a treatment session is activated and terminated, the Arduino board calculates elapsed time and advertises it as a characteristic. When a new treatment session is activated and terminated, the new calculated elapsed time becomes the new advertised characteristic and replaces the old calculated elapsed time. This ensures efficiency and reduces power consumption on the part of the Arduino board, as it does not have to store more than one treatment session’s data at a time. This feature ensures that the CMS can function for many days in a row without researcher intervention, making it useful for the aforementioned clinical trial design.

**Figure 5 bioengineering-11-00502-f005:**
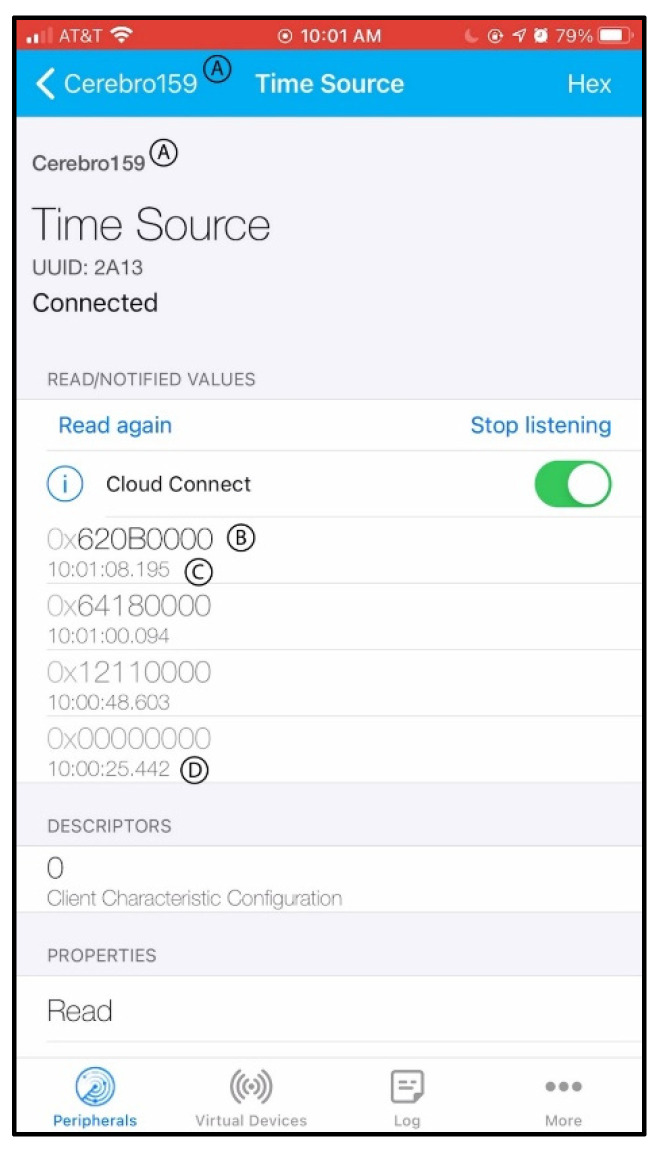
A photo of the graphic user interface (GUI) of the Cerebro Monitoring System (CMS) App. used to establish connection to the CMS and initiate data transmission to the cloud. (A) CMS Pairing ID. (B) Data point transmitted as a hexadecimal (in this case, elapsed time). (C) Time at which the data point is transmitted. (D) Time at which BLE connection between the mobile device and the CMS was established.

**Figure 6 bioengineering-11-00502-f006:**
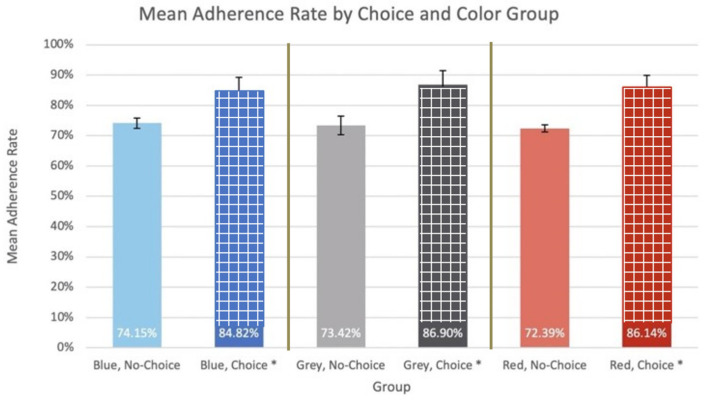
Model 1-mean adherence rate by choice and color group. * Indicates the condition of choice granted and accepted by the patient.

**Figure 7 bioengineering-11-00502-f007:**
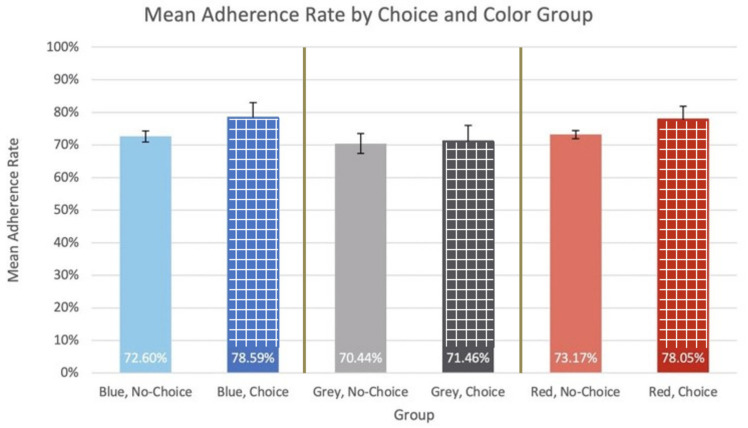
Model 2-mean adherence rate by choice and color group.

**Figure 8 bioengineering-11-00502-f008:**
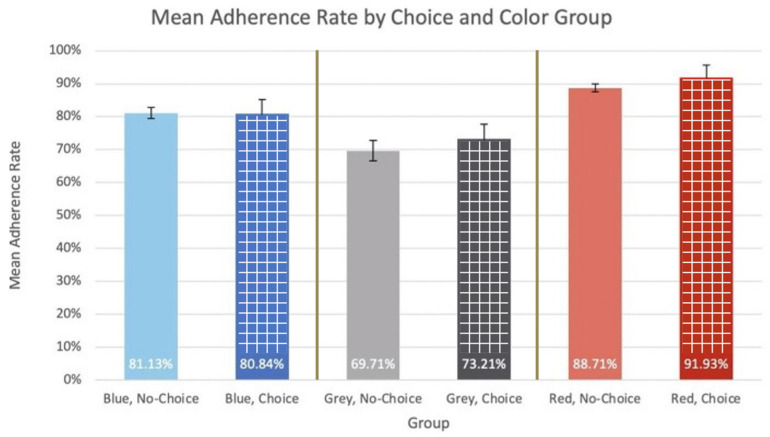
Model 3-mean adherence rate by choice and color group.

## Data Availability

The data presented in this study are available on request from the corresponding author due to privacy considerations.

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
