# Peer review of "Adherence-Promoting Design Features in Pediatric Neurostimulators for ADHD Patients"

_bioengineering, 2024, doi:10.3390/bioengineering11050502_

Round 1

Reviewer 1 Report (Previous Reviewer 1)

Comments and Suggestions for Authors

I find that the authors have fully and comprehensively responded to the comments of all the reviewers and that the proposed work merits publication.

Author Response

We thank the reviewer for their comments.

Reviewer 2 Report (Previous Reviewer 2)

Comments and Suggestions for Authors

The authors have revised their manuscript based on the reviewer's suggestions. I would like to suggest the editors accepting it for publication.

Author Response

We thank the reviewer for their comments.

Reviewer 3 Report (Previous Reviewer 3)

Comments and Suggestions for Authors

This submission bioengineering-2997571 appears to be a resubmission of a previously rejected bioengineering-2895754. In it, Delatte et al. attempted to describe adherence-promoting design features in pediatric neurostimulators for attention deficit hyperactivity disorder (ADHD) patients.

S1. The work explored neurostimulation treatment options for ADHD.

S2. The authors addressed many of the previous concerns.

W1. The authors may consider explicitly listing their contributions of this work.

D1. The similarity rate of bioengineering-2997571 was about 51%, but 42% were similar to Camp's Master's degree thesis (https://oaktrust.library.tamu.edu/bitstream/handle/1969.1/193114/CAMP-THESIS-2021.pdf?isAllowed=y&sequence=1), which can be acceptable for the MDPI.

Author Response

W1. The authors may consider explicitly listing their contributions of this work.

R1. In the present submission we have included the following explicit statement listing our contributions.

Author Contributions

All authors contributed to this work. Conceptualization, A.G-E.; Study design, A.G-E., R.B.K., A.C.; project management, A.C., A.G-E.; hardware and software development, W.D.; data collection and analysis, W.D., A.C., A.G-E.; writing—preparation of the original draft, A.C., A.G-E.; writing—review and editing the manuscript, A.G-E., R.B.K., A.C., W.D.; funding acquisition, A.G-E.. All authors reviewed and approved the publication of this paper.

Reviewer 4 Report (Previous Reviewer 4)

Comments and Suggestions for Authors

This work developed an ADHD adherence improvement device based on an Arduino nano IoT module, it discussed different colors of soft fabric cap monitors for detecting patients' adherence to wearing ADHD therapeutic devices and performed a reasonable statistical analysis, the results show that the color choice helps to improve the patient's adherence. This work is presented completely, the data analysis is clear, and the method of analysis is reasonable. It has good application value. However, there are some issues of lack of clarity, and it is suggested that the authors illustrate the following issues before the work can be accepted.

1, as in the physical picture in Figure 3, there is not a clear show for the specific device mounted on top of the cap, the authors described the devices are hidden in the folds of the cap, which is a bit confusing, it looks like there is not enough space on the caps to hide the Arduino nano module and the battery pack, and the wired connections of the device looks rough and rigid, which also hindered the device attachment, and the overall scale isn't enough to hide it in the folds of the cap. The authors should provide a physical picture with more details of the cap with the mounted sensor device to illustrate how the specific clinical experiment was achieved.

2, the authors describe that the LightBlue@BLE App is used for Bluetooth connection, can the authors provide a screenshot of the mobile device's app when the device is working to show exactly how the device and app works and what data is collected on the app?

3, Figure 4 is not very clear, in PDF normal reading mode, the figure’s font is too small to read and needs to be enlarged, it is suggested that the author replace the current one with a high-resolution figure with a larger font size.

Author Response

This manuscript is a resubmission of an earlier submission. The following is a list of the peer review reports and author responses from that submission.

Round 1

Reviewer 1 Report

Comments and Suggestions for Authors

Adherence-Promoting Design Features in Pediatric Neurostimulators for ADHD Patients

The paper presents an exploration of the intersection between technology and healthcare, in the sector of paediatric ADHD treatment through neurostimulation. The focus on remote health monitoring and at-home care is timely and relevant, given the increasing shift towards these models in contemporary healthcare. The paper highlights a critical challenge in treating ADHD outside of clinical settings: ensuring patient adherence to treatment protocols, which is especially pertinent given the target population's inherent difficulties with task initiation and memory.

The proposed solution, leveraging an Internet-of-Things (IoT) approach for remote adherence monitoring, is innovative and holds significant promise for enhancing treatment outcomes. By examining whether the choice or presence of colour can statistically increase adherence rates among paediatric ADHD patients, the study addresses a nuanced aspect of patient engagement that is often overlooked in clinical trials. The design considerations for the neurostimulation devices, including minimal user input, durability, and accurate adherence data provision, are well-thought-out features that cater to the needs of both patients and healthcare providers.

I think that the paper is great and that it merits publication should a few improvements be met.

The main concern I have regards the reproducibility of the method. The paper discusses extensively the analysis of the acquired psychophysical data. However, it does not provide enough technical details for reproducing the algorithms running on the IoT Arduino device. The technical specifications provided are of secondary importance and can be looked up. What I find missing is a description of the algorithm that runs on this device and implementation remarks that would help the reader recreate the proposed work.

The rest of the comments are of secondary important and regard the completeness and the improvement of the presentation of the proposed work.

However, the paper could benefit from a brief discussion of the theoretical framework or preliminary findings to provide a stronger foundation for the study's significance. Additionally, the integration of patient and physician feedback into the development process, if any, would be valuable to understand the user-centred design approach further.

Given that the study focuses on a very specific population (paediatric ADHD patients) and a specific treatment modality (neurostimulation), a reviewer might raise concerns about the generalizability of the findings to other populations or other forms of treatment.

While the use of IoT for remote adherence monitoring is innovative, concerns might be raised about the reliability of the technology, potential technical issues, and how accessible such technology would be to various socioeconomic groups. What measures are in place to ensure consistent and accurate data collection?

The use of remote monitoring technologies, especially in a paediatric population, raises significant ethical considerations regarding privacy, consent, and the potential for over-surveillance. How are these concerns addressed in the study design?

The paper mentions minimal user input as a feature of the device, but what is the applicability of the method in children in terms of user friendliness and acceptance.

The paper claims that the technology can increase assurance of effective at-home care, but what about evidence or metrics that demonstrate the clinical significance and real-world impact of improved adherence through this intervention?

There might be a lack of comparison with existing adherence strategies or technologies. Is there any room or relevant works that could discuss this work comparatively analysis to demonstrate the superiority or added value of the proposed approach?

Reviewer 2 Report

Comments and Suggestions for Authors

Major concern

The major problem of this study is the lack of controlling for participants’ motivation to receive the neurostimulation treatment for ADHD, severity of ADHD symptoms, and the effect of neurostimulation. All factors influence participants’ medical adherence.

Minor concern

Title: “ADHD” should be changed into the full spelling.

Line 58: A full spelling for “ADHD” should be added.

Line 76: A full spelling for “FDA” should be added.

Line 77: [16, 17] [18].

Line 162: “Cerebro Monitoring System (CMS)” The abbreviation has been introduced on line 95.

Reviewer 3 Report

Comments and Suggestions for Authors

In bioengineering-2895754, Camp et al. attempted to describe adherence-promoting design features in pediatric neurostimulators for attention deficit hyperactivity disorder (ADHD) patients.

S1. The work explored neurostimulation treatment options for ADHD.

W1. Presentation can be improved.
W2. The authors did not explicitly list their contributions (if any) of this work.
W3. Formatting was unusual (esp. for equations).
W4. There are inconsistent citation styles in the main text (e.g., "[8, 13, 14]", vs. " [16, 17] [18]").
W5. Details of clinical trials were not clearly described.
W6. It is unclear how many patients aged 8-12 were participated.
W7. Results on all three models showed either color choice or color was not significant. "More work should be done."
W8. The work did not appear to be conclusive.
W9. The authors did not compare with related work.

Reviewer 4 Report

Comments and Suggestions for Authors

In this study, the authors developed a device that monitors the time of pediatric patients with ADHD wearing electrical stimulation equipment, the device that is claimed to improve patient adherence. In concept alone, this design makes excellent clinical sense and has good practical value for improving patient adherence. If it is integrated into existing ADHD stimulation equipment, the device could potentially improve treatment outcomes.

However, after reading the entire article, I am confused about the results that the authors didn’t provide real clinical validation for the device. The authors did not provide specific information about the ADHD equipment and the participants but only gave three models with different conclusions, which are more like randomized results generated by simulation than actual results.

This cannot validate whether the proposed IOT device is practically feasible. If the authors can provide more detailed information about the clinical trials, this article is worthy of being accepted. If the authors only processed a simple signaling transmission device on an Arduino platform and proposed some hypothetical application scenarios, this work is not well accomplished, and it is recommended that the authors resubmit the article after adding sufficient clinical trials.